

# The pyrogeography of eastern boreal Canada from 1901 to 2012 simulated with the LPJ-LMfire model

Emeline Chaste[1,2], Martin P. Girardin[1,3], Jed O. Kaplan[4], Jeanne Portier[1], Yves Bergeron[1,5], Christelle Hély[2,5]

[1]Département des Sciences Biologiques, Université du Québec à Montréal and Centre for Forest Research, Case postale 8888, Succursale Centre-ville Montréal, QC H3C 3P8, Canada
[2]EPHE, PSL Research University, ISEM, Univ. Montpellier, CNRS, IRD, CIRAD, INRAP, UMR 5554, F-34095 Montpellier, FRANCE
[3]Natural Resources Canada, Canadian Forest Service, Laurentian Forestry Centre, 1055 du PEPS, P.O. Box 10380, Stn. Sainte-Foy, Québec, QC G1V 4C7, Canada
[4]Institute of Earth Surface Dynamics, University of Lausanne, Geopolis, Quartier Mouline, 1015 Lausanne, Switzerland
[5]Forest Research Institute, Université du Québec en Abitibi-Témiscamingue, 445 boul. de l'Université, Rouyn-Noranda, QC J9X 5E4, Canada

*Correspondence to*: Emeline Chaste (emeline.chaste@canada.ca)

**Abstract.** Wildland fires are the main natural disturbance shaping forest structure and composition in eastern boreal Canada. On average, more than 700,000 ha of forest burns annually, and causes as much as C\$2.9 million worth of damage. Although we know that occurrence of fires depends upon the coincidence of favourable conditions for fire ignition, propagation and fuel availability, the interplay between these three drivers in shaping spatiotemporal patterns of fires in eastern Canada remains to be evaluated. The goal of this study was to reconstruct the spatiotemporal patterns of fire activity during the last century in eastern Canada's boreal forest as a function of changes in lightning ignition, climate and vegetation. We addressed this objective using the dynamic global vegetation model LPJ-LMfire, which we parametrized for four Plant Functional Types (PFTs) that correspond to the prevalent tree genera in eastern boreal Canada (*Picea, Abies, Pinus, Populus*). LPJ-LMfire was run with a monthly time-step from 1901 to 2012 on a 10-km$^2$ resolution grid covering the boreal forest from Manitoba to Newfoundland. Outputs of LPJ-LMfire were analyzed in terms of fire frequency, net primary productivity (NPP), and aboveground biomass. The predictive skills of LPJ-LMfire were examined by comparing our simulations of annual burn rates and biomass with independent datasets. The simulation adequately reproduced the latitudinal gradient in fire frequency in Manitoba and the longitudinal gradient from Manitoba towards southern Ontario, as well as the temporal patterns present in independent fire histories. Nevertheless, the simulation led to underestimation and overestimation of the fire frequency at both the northern and southern limits of the boreal forest in Quebec. The general pattern of simulated total tree biomass also agreed well with observations, with the notable exception of overestimated biomass at the northern treeline, mainly for *Picea* PFT. In these northern areas, the predictive ability of LPJ-LMfire is likely being affected by a low density of weather stations, which has led to underestimation of the strength of fire-weather interactions during extreme fire years and, therefore, vegetation consumption. Agreement of the spatiotemporal patterns of fire frequency with the observed data confirmed that fire in the study area is strongly ignition-limited. Overall, climate and lightning ignition variability at multi-decadal and -annual time-





scales was the primary driver of fire activity since the beginning of the 20th C. However, our simulations highlighted the importance of both climate and vegetation on fire: despite an overarching $CO_2$-induced enhancement of NPP in LPJ-LMfire, forest biomass was relatively stable because of compensatory effects of increasing fire activity.

## 1 Introduction

Wildland fires are the main natural disturbance shaping forest structure and composition in eastern boreal Canada (Bergeron et al., 1998, 2014). On average, more than 0.7 Mha burn annually across Manitoba, Ontario, Quebec and the Maritime Provinces, and causes as much as \$C2.9 million worth of damage and property losses (Canadian Council of Forest Ministers, 2017). About 97 % of these burned areas are generated by a small proportion (3 %) of large fires (fires > 200 ha in area; Stocks et al., 2003). For example, a fire burned 583,000 ha within a few days in 2013 near the aboriginal community of Eastmain

(Province of Quebec), which is the equivalent of 31 % of the total area burned during that year in Quebec (Erni et al., 2017). Studies of the spatial distribution of wildland fires in the past have highlighted that the frequency and size of fires in Canada have continuously increased over the last 50 years or so in response to ongoing global warming (e.g., Kasischke and Turetsky, 2006; Hessl, 2011; Girardin and Terrier, 2015). Concerns are now being raised about the increasing frequency/severity of extreme climatic events with further warming, which could lead to an increasing concentration of numerous large fires in time

and space (Wang et al., 2015). Given these observations and projections, there is growing concern about the capacity of the boreal forest to recover from disturbances (Bond et al., 2004; IPCC, 2013; Kurz et al., 2013; Rogers et al., 2013).

Wildland fire regimes are described by several attributes including the frequency, size, intensity, seasonality, type and severity of fires (Keeley, 2009). The spatiotemporal variability of a fire regime depends upon the coincidence of favourable conditions for fire ignition, fire propagation and fuel availability, which are controlled by ignition agents, weather/climate, and

vegetation (Flannigan et al., 2009; Moritz et al., 2010). Almost half of the fires that occur in eastern boreal Canada are ignited by lightning and represent 81 % of the total area burned (Canadian Forest Service, 2016), while the remaining fires originate from human activities. The capacity of a fire to grow into a large fire is determined by many factors, which include weather and fuel. High temperature, low precipitation, high wind velocity and low atmospheric humidity can increase the growth of these fires (Flannigan et al., 2000). The intensity, severity and size of fires are further influenced by the species composition

within a landscape, with needle-leaf species being more fire-prone than broad-leaf species owing to their high flammability (Hély et al., 2001). Physical variables such as slope, surficial deposits and soil moisture can also have significant effects on the rate at which fires spread by influencing fuel moisture or creating natural fire breaks (Hély et al., 2001; Mansuy et al., 2011; Hantson et al., 2016). Climate change scenarios for Canada indicate an increase in both temperature and precipitation in the coming decades. Yet, the increase in precipitation should not compensate for the increase in temperature (IPCC, 2013),

and a greater moisture deficit is expected compared to the current state. Warmer springs and winters that lead to an earlier start of the fire season are anticipated, together with an increase in the frequency of extreme drought years due to more frequent and persistent high-pressure blocking systems (Girardin and Mudelsee, 2008). These phenomena are expected to lead to an





increase in the frequency and size of fires in eastern boreal Canada in response to ongoing global warming (Ali et al., 2012). Effects of these changes in seasonal onset and dryness are such that the average size of spring wildfires could be multiplied by a factor of three for each additional 1°C of warming (Ali et al., 2012; Girardin et al., 2013a; Price et al., 2013). An increase in the areas burned would affect both plans for forest management and fire suppression strategies. It could also have subsequent

feedbacks on the global carbon cycle, given that the substantial quantities of carbon currently being stored in these landscapes could be re-emitted back into the atmosphere (Pan et al., 2011).

A number of uncertainties persist concerning future fire trajectories and biases still exist regarding our current understanding of the natural variability of fire regimes. Forest management and fire suppression since the 1970s have contributed to modifying fire patterns and vegetation attributes in Canada (Gauthier et al., 2014). Climate also has been rapidly

changing in recent decades with the expansion of human activities. All of these changes have altered the interactions between fire regimes and their various controls (Bergeron et al., 2004b). For instance, a lack of quantitative knowledge regarding the relative changes in fire regimes that are attributable to human impacts prevents modellers from adequately taking into account their influence in future fire trajectories. Although it is also possible to investigate recent changes in fire regimes that are based upon observations collected over relatively short time intervals (< 100 years), reliable observations were not available in many

boreal regions prior to the late 1960s (Podur et al., 2002). Furthermore, fire history studies rarely consider the feedbacks of fires on vegetation, mostly because historical data about vegetation composition are lacking (Danneyrolles et al., 2016). This is particularly the case for studies dealing with reconstructions of fire activity using dendrochronological evidence (e.g., Girardin et al., 2006) or by adjusting empirical datasets (Van Wagner et al., 1987). This problem may be circumvented by investigating past fire regimes over long periods of time through the analysis of charcoal and pollen in soil layers or lacustrine

deposits (Payette et al., 2008; Ali et al., 2009). However, these paleoecological methods are costly, time-consuming and incapable of capturing the overall spatial variability of fire regimes at annual- to decadal-scale resolutions. Faced with these gaps, increasing our knowledge of the spatiotemporal patterns of past fires is necessary to perform better predictions in the future.

Simulations using dynamic global vegetation models (DGVMs) allow the spatiotemporal distribution of fires to be

estimated relative to climate and vegetation (Yue et al., 2015; Hantson et al., 2016). Indeed, these models simulate shifts in potential vegetation composition and related fire activity in response to changes in climate or environmental constraints (Smith et al., 2001). Experiments can be conducted at fine- to broad-spatial scales and validated at relatively short- to medium-timescales. Validation can be realized in regions where human activities are sufficiently low to allow comparisons with natural potential vegetation, by comparing simulation results with high-resolution satellite products at global scales, such as MODIS

(Tang et al., 2010). DGVM simulations also may be validated at decadal- to millennial-timescales by comparing them with historical records of vegetation or fire activity that have been reconstructed using indicators derived from pollen and charcoal, amongst others, which are deposited in lacustrine sediments (Molinari et al., 2013). One of these models, the Lund-Postdam-Jena (LPJ) model has been the subject of numerous refinements over time, especially in simulations of fire patterns (Thonicke



et al., 2010; Pfeiffer and Kaplan, 2012), and has been validated in many regions worldwide, excluding eastern boreal Canada for example (e.g., Prentice et al., 2011; Pfeiffer et al., 2013; Yue et al., 2015; Knorr et al., 2016).

Here, we used the LPJ-LMfire model that was developed by Pfeiffer et al. (2013) to perform a simulation experiment that targeted the boreal forest of eastern Canada and which covered the last century, with customized parametrization to capture for prevalent tree genera present in eastern boreal Canada. The DGVM explicitly simulates fire ignition from lightning and, hence, is particularly adapted to the largely ignition-limited fire regimes in our study region. The objectives of this study were (1) to calibrate the LPJ-LMfire model for boreal forests in eastern Canada, (2) to verify the predictive skills of the model with independent datasets from eastern Canada's boreal forests, (3) to reconstruct fire activity, net primary productivity and aboveground biomass during the last century, and (4) to determine how the spatiotemporal pattern of these three components have evolved conjointly in relation to changes in climate variables.

## 2 Model, Experimental Setup and Methods

### 2.1 Study area

The study area encompasses eastern Canada's boreal forest (Brandt, 2009) from Manitoba to Newfoundland, which ranges from 102.86°W to 52.64°W and from 46.61°N to 64.71°N (Fig. 1). The most common needle-leaf tree species present in this region are black spruce (*Picea mariana* (Mill.) BSP), white spruce (*Picea glauca* (Moench) Voss), balsam fir (*Abies balsamea* (L.) Mill.), jack pine (*Pinus banksiana* Lamb.), white pine (*Pinus strobus* L.), red pine (*Pinus resinosa* Sol. ex Aiton), eastern larch (*Larix laricina* [Du Roi] K.Koch), and eastern white cedar (*Thuja occidentalis* L.). The main broad-leaf tree species are trembling aspen (*Populus tremuloides* Michx.) and white or paper birch (*Betula papyrifera* Marsh.) (Ecological Stratification Working Group, 1996; Brandt, 2009; Shorohova et al., 2011). The study area is divided from south to north into four ecozones (Fig. 1; Ecological Stratification Working Group, 1996). 1) The Boreal Shield ecozone is characterized by rocky and rugged landscapes influenced by a continental climate (long and cold winters; short and warm summers) and by the cold air masses flowing out from Hudson Bay. Fire frequency is typically high. Landscapes are dominated by needle-leaf tree species in the westernmost areas and codominance of deciduous tree species in temperate eastern areas. 2) The Boreal Plain ecozone corresponds to drier areas that are characterized by glacial deposits of variable thickness on flat or slightly rolling terrain. Forests are dominated by mixed boreal species, mainly represented by black spruce, trembling aspen and jack pine. 3) The Hudson Plain ecozone is characterized by sparser vegetation, which is dominated by *Sphagnum* and shrubs. Poor drainage conditions constrain southern trees to establish at drier, higher elevations. 4) The Taiga Shield ecozone, which is split into Eastern and Western parts, is characterized by colder climate conditions. The landscape becomes more open along a latitudinal gradient from south to north. Dominant tree species include black spruce and jack pine.




### 2.2 LPJ-LMfire model

Simulations of the terrestrial ecosystem were carried out using the dynamic global vegetation model LPJ-LMfire, which contains updates of both LPJ and the SPread and InTensity of the FIRE (SPITFIRE) wildfire module (Thonicke et al., 2010). The model has been most extensively evaluated for boreal forest (Pfeiffer et al., 2013). LPJ-LMfire is designed to simulate

regional ecosystem dynamics, structure and composition, with vegetation and fire events as responses to changes in climate and carbon dioxide concentration ($CO_2$) (Sitch et al., 2003). LPJ-LMfire describes the state of an ecosystem in terms of annual carbon stocks (living biomass, litter and soil), Net Primary Productivity (NPP), net biome productivity, evapotranspiration, heterotrophic respiration, soil moisture fraction, and forest structure and vertical profile (cover fraction, individual density, crown area, leaf area index). In the present study, changes in vegetation state are described in terms of NPP and total carbon

stocks in the living aboveground biomass. In LPJ-LMfire, vegetation is defined by up to nine Plant Functional Types (PFTs). Each PFT represents one or several species sharing the same physiology and dynamics, governed by a short list of vital attributes, and constrained by bioclimatic limits (Sitch et al., 2003). Vegetation dynamics are updated annually based on simulation of daily and annual processes. Daily processes were defined in terms of photosynthesis, stomatal regulation, soil hydrology, autotrophic respiration, leaf and root phenology and decomposition. Annual processes were defined in terms of

several sources of mortality, seedling establishment, reproduction, allocation and tissue turnover (Smith et al., 2001; Sitch et al., 2003). The computational core of SPITFIRE is based upon Rothermel-type surface fire behaviour models (Rothermel, 1972; Andrews et al., 2008) and is designed to simulate processes of natural fires and their implications on vegetation mortality and fire emissions (Thonicke et al., 2010). The LMfire module simulates lightning ignitions based upon a daily time-step, and uses fuel bulk density and fuel moisture to calculate the fire rate of spread, intensity, and fire mortality. It allows fires to burn

over multiple days and simulates fire extinction from changes in weather and fuel (Pfeiffer et al., 2013).

### 2.3 Simulation protocol

LPJ-LMfire was run monthly from 1901 to 2012 on a 10 × 10 km equal-area grid covering eastern boreal Canada from Manitoba to Newfoundland. Driver datasets were prepared in netCDF format and are described in Table 1. Climate data were compiled at a monthly time-step, while atmospheric $CO_2$ concentrations ($[CO_2]$) were compiled at an annual time-step (see

Section 2.4). A 1120-year spinup period was prescribed to equilibrate vegetation and carbon pools with climate at the beginning of the study period (Smith et al., 2001), and to ensure that forest biomass was in stable condition with fire disturbances (Tang et al., 2010). This spinup run was made using linearly detrended 1901-2012 climate data, repeated 10 times.

### 2.4 Environmental input data sets

### 2.4.1 Climate

Monthly means of temperature, diurnal temperature range, precipitation, number of days with precipitation, and wind speed were extracted between 1901-2012 from Environment Canada's historical climate database (Environment Canada, 2013) using





BioSIM software (v.10.3.2; Régnière et al., 2014). Gridded climate data were prepared in BioSIM by interpolating weather data from the four weather stations that were closest to each $10 \times 10$ km grid, adjusted for elevation and location differentials with regional gradients, and averaged using inverse distance weighting ($1/d^2$, where $d$ is distance). Missing wind speed values between 1901 and 1968, and those for 2010-2012 were set to the monthly 1969−2010 averages.

Monthly means of total cloud cover percentage for the entire atmosphere and convective available potential energy (CAPE) were interpolated on our grid from the NOAA-CIRES 20[th] Century Reanalysis v2 dataset at ~2.0° latitude and 1.75° longitude resolution (Compo et al., 2011). For a given grid cell, the annual monthly CAPE anomaly was calculated as the difference between the annual value and the monthly normal for CAPE, which was computed between 1961 and 1990.

### 2.4.2 Lightning

The Canadian lightning detection network (CLDN) dataset, covering the period 1999-2010 (Orville et al., 2011), was used to reconstruct the monthly cloud-to-ground lightning strike density (number/day/km$^2$) between 1901 to 2012. Given the strong correlation between lightning strikes and the product of CAPE and precipitation (e.g., Peterson et al., 2010; Romps et al., 2014), we computed daily strike density using CAPE data and distributed the lightning strikes over the daily fraction of monthly rainy days (Pfeiffer et al., 2013). Across Canada and within our study area, July was the month with the maximum

number of strikes between 1999 and 2010 (supplement S1 Fig. S1 A), and in turn, inter-annual strike variability (hereafter, referred to as min-to-mean and max-to-mean ratios) ranged from 0.1 to 7.5 times the July mean (supplement S1 Fig. S1 B). This inter-annual variability in lightning strikes was preserved in our reconstruction by applying these two ratios to normalize values between −1 and +1 of CAPE anomalies (following Pfeiffer et al., 2013; see supplement S1 for further details), and were directly added to the 1999−2010 flash climatology.

### 2.4.3 Soils

The volume fraction of coarse fragments, together with the 0−100 cm depth soil texture fractions of sand and clay, were interpolated on the $10 \times 10$ km grid from the 1 km resolution ISRIC – World Soil Information dataset (Hengl et al., 2014).

For topography, we interpolated the 30 arc-second gridded digital elevation model (DEM) of Canada (Natural Resources Canada, 2007). We calculated slopes in degrees at 30 arc-seconds with the DEM map using ArcGIS 10.4.1 and interpolated

the data to our 10 km grid. To calculate the percentage of land (i.e., removing lakes and water course areas) in each grid cell, we rasterized the water fraction of the National Hydro Network (NHN) dataset at 100 m resolution (Natural Resources Canada, 2010). We calculated water fraction at 10 km resolution from grid cells at 100 m resolution that had a percentage of water fraction > 50 %. The land fraction was defined as the inverse of the water fraction. Roads, power lines, dams, mines, and other human-made structures, and areas of bare rock, were not considered in this study.



### 2.4.4 Atmospheric CO₂ concentration

Monthly mean atmospheric carbon dioxide ($CO_2$) concentrations covering the periods 1901 to 1980 and 1981 to 2012 were obtained from Pfeiffer et al. (2013) and the Mauna Loa dataset (Keeling et al., 2009), respectively. Annual mean atmospheric $CO_2$ varied from 296.23 ppm in 1901 to 392.48 ppm in 2012, which corresponds to an increase of 32.5 %.

5    **2.5 PFT definitions and LPJ-LMfire model modifications**

LPJ-LMfire was calibrated for four Plant Functional Types (PFTs) that corresponded to the predominant tree genera currently present in the boreal forest of Canada: *Picea, Abies, Pinus* and *Populus*. PFT-related parameters, e.g., fraction of roots in upper soil layer or minimum and maximum temperatures of the coldest month for establishment, were assigned values from the published literature or global databases (see supplement S2 for further details).

10    **2.5.1 Edaphic limits to establishment**

Establishment and growth of boreal tree species are influenced by a wide range of soil properties that were related to soil nutrient availability, and which included pH, parent material, soil particle size, and water content, among others (Girardin et al., 2001; Beauregard and de Blois, 2014; Gewehr et al., 2014). Not all ecosystem processes linking these properties to tree establishment are simulated in the current version of LPJ-LMfire. Notably, the model does not simulate the development of 15    peatlands or the process of paludification, and does not include a complete module of biogeochemical cycling in soils that would emulate processes leading to acidification, for instance. As has been proposed by Beauregard and de Blois (2014), however, some edaphic variables may be indicative of certain soil processes at the stand level. In this study, correlations between the abundance of specific tree genera and soil clay content led to the implementation of a simple scheme to limit tree establishment in LPJ-LMfire (supplement S3 Fig. S2 A). Edaphic limits to establishment were defined here in the same way 20    that bioclimatic limits are used in LPJ. The correlations between genus-specific tree cover fraction from Beaudoin et al. (2014) and clay volume fraction from Hengl et al. (2014) were analyzed at 10 km resolution. For each PFT, the percentage of clay corresponding to the upper limit of the 90 % CI of its distribution, for grid cells with at least 10 % of PFT cover, was used in the model as a threshold, above which the given PFT could not establish. The upper limit of the 90 % CI of the clay percentage distribution was 20 %, 13 %, 18 % and 23 % for *Picea*, *Abies*, *Pinus* and *Populus*, respectively (supplement S3 Fig. S2 A). 25    The 20 % threshold essentially results in exclusion of *Picea* and *Populus* PFTs in the Hudson Plain (supplement S3 Fig. S2 B and C), while the threshold of 13 % leads to additional exclusion of other PFTs, especially *Pinus*, in the Midwestern Boreal Shield and Boreal Plain (supplement S3 Fig. S2 B and C).

### 2.5.2 Post-fire recruitment

Recruitment of *Pinus banksiana* requires the heat of fires to release seeds from serotinous cones (Gauthier et al., 1996). This 30    condition was implemented in the current LPJ-LMfire version specifically for the *Pinus* PFT by inhibiting seedling





establishment during years without fire. Such fire effects on seed dispersal are also observed for *Picea mariana*, which has semi-serotinous cones. Given that black spruce cones can open gradually over time in the absence of fire (Messaoud et al., 2007), *Picea* PFT establishment was not constrained by fire occurrence. Establishment of the *Abies* and *Populus* PFTs are also not constrained by fire occurrence.

## 2.6 Model evaluation

We assessed the performance of our customized LPJ-LMfire by comparing simulation results with previously published datasets on fire and maps of genus-specific aboveground biomass for Canada's forests.

### 2.6.1 Fire activity

The simulated burned area fraction was evaluated against three fire data products. First, annual burn rates during 1980−2012 were compiled from the Natural Resources Canada fire database (M.A. Parisien, personal communication) using the Canada's national fire polygons with the hexagonal cells approach from Héon et al. (2014), but extended to our study area. We used 365 hexagonal cells to cover our study area and to compute the 1980 to 2012 simulated mean annual burn rates with 95 % confidence intervals (95 % CI) for each hexagonal cell. The second fire data product originated from stand-replacing fire history studies. Here, historical annual proportions of burned areas were obtained for 26 locations (supplement S4 Fig. S3) using post-fire stand initiation reconstructions based upon field and archival data that were digitized and which were included in GIS databases (Girardin et al., 2013b; Héon et al., 2014; Portier et al., 2016). Using a 100-km radius around each location centroid, we calculated the simulated mean annual burn rates between 1911 and 2012, together with the 95 % CI. Differences between our simulated estimate 95 % CIs and these two fire data products were considered qualitatively as 'not different' if the observed annual burn rate fell within the 95 % CI of the simulated mean burn rate. Note that as the period covered by the historical fire data often extended further back in time into the 19th or 18th centuries for southern locations (supplement S4 Table S2), some important differences could be expected in the comparison process. Finally, a third validation of fire simulations was made by comparing time-series of total simulated annual burned areas in boreal forests of Manitoba, Ontario and Quebec with provincial fire statistics (point data) from the Canadian National Fire Database (CNFDB; Canadian Forest Service, 2016) covering the 1959-2012 period. Human-caused fires were excluded from these analyses. Spearman rank correlation ($r_s$) was used to quantify the agreement between observed and simulated data. The agreement between simulation and observation was further evaluated in terms of fire seasonality by comparing their respective distributions of mean monthly areas that were burned from 1959 to 2012.

### 2.6.2 Aboveground biomass

Published maps of total aboveground biomass at the genus level (Beaudoin et al., 2014) were used to evaluate model simulations. Maps that were created by Beaudoin et al. (2014) were constructed at 250-m spatial resolution using remote sensing MODIS datasets, combined with photo-plot observations of Canada's National Forest Inventory (NFI), mainly in the





southern areas (see non-hatched area on Fig. 4). We aggregated the 250-m data to a 10-km resolution and applied a correction for the vegetated treed fraction of the landscape, as defined by Beaudoin et al. (2014). Vegetated treed fraction corresponds to the fraction of the grid cells that are covered by tree species of any size on at least 10 % of the grid cell.

Total aboveground biomass that was estimated by two other methods reported by Margolis et al. (2015) was used for a second evaluation of model simulations for the 5 ecozones under study. The Boreal Shield ecozone was divided into three ecoregions for comparison (Fig. 1); ecoregions corresponding to the classification of ecological regions on a finer scale than ecozones. The first method of biomass estimation is based upon the 'Geosciences Lidar Altimetry System' (GLAS) method, which estimates total aboveground biomass from the waveforms recorded over vegetated land using Lidar instruments. The second method is based upon NFI photo-plot estimates of total aboveground biomass using allometric equations.

## 2.7 History of the eastern boreal forest of Canada described by LPJ-LMfire

The outputs of LPJ-LMfire for the eastern boreal forest of Canada were analyzed in terms of annual burn rates, NPP and total aboveground biomass. Significant changes in each temporal series were highlighted by a regime shift calculation developed by Rodionov (2004, 2006). A sequential application of Student's $t$-test on 1,000 randomly chosen grid cells was used (Rodionov, 2004, 2006). To be statistically significant at $P = 0.10$, the difference ($diff$) between mean values of two subsequent periods that was determined according to Student's $t$-test should satisfy the condition:

$$[1] \quad diff = t\sqrt{2\sigma_i^2 / l},$$

where $t$ is the value from the $t$-distribution with $2l - 2$ degrees of freedom at the given probability level $P$, $l$ is the cut-off length of the growth phase to be determined (hereafter, set to periods of 20 years), and $\sigma_i^2$ is the average variance for running l-year intervals. The sample proportion, representing the fraction of $k$ cells (an integer $\geq 0$) of a given population N (an integer $> 0$), which was identified positively as recording a growth decline (or release), a biomass reduction (or biomass increase) and an increase of fire activity (or decrease), was computed for each sampled year from 1920 to 2007.

## 2.8 Sensitivity analysis to CO₂ fertilization

In terrestrial ecosystem models, changes in atmospheric [$CO_2$] in the recent past and future often have a more important influence on vegetation than does climate change (Girardin et al., 2011). Therefore, their inclusion has a very important effect on simulated changes in productivity. Here, the effect of [$CO_2$] fertilization was explored using two simulations. In the first simulation, we ran the model with a constant [$CO_2$] from 1901 to 2012, which was fixed at 296.23 ppm (year 1901 value). In this case, there was no response of vegetation gross primary production (GPP) or fire to changes in [$CO_2$], i.e., 'Climate-only' experiment. In the second simulation, we ran the model with increases in the [$CO_2$] ('Climate + $CO_2$' experiment). For further details, see section 2.4.4. The effect of $CO_2$ fertilization on vegetation and fire is determined by the difference between simulations 'Climate + $CO_2$' and 'Climate-only.'



## 3 Results

We report on the evaluation of the process-based model performance in adequately simulating spatial patterns of fire frequency and fuel conditions (as indicated by the aboveground biomass of the four PFTs and total net primary productivity, NPP) in eastern boreal Canada. We also report changes in the fire activity during the last century simulated by LPJ-LMfire, with
associated changes in vegetation features.

### 3.1 Predictive skills of the LPJ-LMfire model

#### 3.1.1 Fire activity

For the recent period 1980-2012, mean and maximum simulated annual burn rates were respectively 0.36 % yr$^{-1}$ and 1.49 % yr$^{-1}$ (Fig. 2 B), while the mean and maximum observed annual burn rates were 0.28 % yr$^{-1}$ and 2.03 % yr$^{-1}$ (Fig. 2 A). Observed
and simulated burn rates were not significantly different in more than 80 % of the studied hexagonal cells (295 of 365; Fig. 2 C). LPJ-LMfire was able, therefore, to capture the amplitude of interregional variation. Decreases in fire activity observed along both the latitudinal gradient in Manitoba, and the longitudinal gradient from Manitoba to southern Ontario were well reproduced by the simulation (Fig. 2 A and B). Furthermore, more than half of the observed historical annual burn rates fell within the 95 % CI of their corresponding simulated annual burn rates (for further details, see supplementary S4 Table S2 and
Fig. S3). LPJ-LMfire overestimated annual burn rates from south of Hudson Bay in Ontario to southwestern Quebec (Fig. 2 C), while it underestimated the annual burn rates in the western areas of the central boreal forest in Quebec (Fig. 2 C). Spearman correlation coefficients ($r_s$) of time-series of observed versus simulated areas burned are 0.41 for Quebec and 0.50 for Ontario and Manitoba (Fig. 3 A). As revealed by these coefficients, LPJ-LMfire was also able to emulate year-to-year variability in annual areas that were burned in Manitoba and Ontario, but less so in Quebec. High fire activity years over the temporal series
were also captured in the simulations, including 1961, 1968, 2003 and 2005, mostly in Manitoba and Ontario (Fig. 3 A). However, three extreme fire years were not reproduced: 1983, 1989 and 2002 (Fig. 3 A). Based upon the comparison of monthly percentage of total areas that were burned between 1959 and 2012 in eastern boreal Canada, the simulated fire season generally started one month earlier than was observed (Fig. 3 B).

#### 3.1.2 Fuels

Overall, the general latitudinal pattern of simulated total tree biomass (Fig. 4 A2) agreed with the pattern of observed total tree biomass (Fig. 4 A1). Median simulated total tree biomass (with 90 % CI) in the southern areas (non-hatched) was 77 T ha$^{-1}$ (33-108 T ha$^{-1}$), while median observed total tree biomass in the same areas was 73 T ha$^{-1}$ (36-100 T ha$^{-1}$). In the Boreal Shield ecozone, percentage differences between mean total tree biomass that was simulated and that which was estimated by NFI-based and GLAS-based methods were respectively 31 % and -7.8 %, and decreased along a westward gradient from Quebec
to Manitoba (Table 2). We greatly overestimated mean total tree biomass in the Boreal Plain ecozone because these differences were -60 % and -50 %, respectively. For the Taiga Shield ecozone in Quebec and Manitoba that corresponds to less intensively



sampled northern regions (hatched areas), total tree biomass was largely overestimated, mostly in Quebec due to high genus-specific biomass of the *Picea* PFT (Fig. 4 B). In this ecozone, relative differences with GLAS-based estimates ranged from 1.3 % in west to 63.6 % in east, whereas it was only 1.6 % in comparison with NFI-based estimates (Table 2). Greater relative differences were observed in the Hudson Plain ecozone (Table 2), where we overestimated total tree biomass for grid cells in which edaphic limits were not too restrictive and where vegetation could establish (Fig. 4 A2). This overestimation was mainly due to high biomass of *Picea* and *Populus* PFTs (Fig. 4 B). Despite scale-local overestimates, the range of genus-specific biomass variability of *Abies* and *Populus* PFTs was well captured.

## 3.2 Fire history simulated by LPJ-LMfire

### 3.2.1 Fire activity

Simulated burn rates displayed multi-decadal variation over the 20th C., notably in Manitoba and Ontario (Fig. 5 A). High fire activity that was reported in the period 1910-1930 was followed by a decrease in fire activity until the 1970s, and then increased to levels similar to those of the early 20th C. (Fig. 5 A). Since the 1970s, annual burn rates increased in central Manitoba and western Ontario, and south-central areas of Quebec (Fig. 5 A). Episodes of successive years of intense fire activity occurred in 1908-1910, 1919-1923, 1995-1998 and 2002-2007 (supplement S6 Fig. S5 A). The simulated fire season was not stationary: a fire seasonality index (FSI) was computed as the percentage difference between spring and summer total burned areas (supplement S6 Fig. S5 B), and varied between 0.17 % and 83 %. The period from the end of the 1960s to end of the 1990s corresponded to a period during which several years of high FSI were observed compared to the entire time series. FSI that was greater than 50 % was calculated for 1968, 1977, 1980 and 1993 (supplement S6 Fig. S5 B). May and June were consistently the spring months with the largest burned areas, while summer months recorded fewer and fewer burned areas over the course of the 20th C.

### 3.2.2 Fuels

For the 'Climate + $CO_2$' experiment, simulated annual NPP that was averaged over the entire study region and whole period was 5.4 T ha$^{-1}$, with a minimum of 4.2 T ha$^{-1}$ in 1907 and a maximum of 7.1 T ha$^{-1}$ in 2003 (Fig. 5 B). Both sequential *t*-test analysis and temporal time-series showed that NPP increased since the 1970s (Fig. 6 A and B), largely in southern areas of Quebec and in eastern Ontario (Fig. 5 B). This constant increase in NPP since the 1970s was not observed in Manitoba and western Ontario, where a significant increase in annual burn rates was observed (Fig. 5 A). Some regions in south-central Ontario showed a decline in NPP during the early 20th C. and the same trend was observed in south-central Quebec since the 1980s. The proportion of cells recording a decline of NPP was particularly noteworthy in 2004 and 2006 (Fig. 6 A and B).

Annual simulated NPP, averaged over the whole area, was positively correlated with annual atmospheric [$CO_2$] ($r^2 = 0.767$, $P < 0.001$). Mean percentage increase in NPP that incurred by rising [$CO_2$] for our 5 periods was 2.7 %, 5.5 %, 8.9 %, 16.7 %





and 27.6 % (supplement S7 Fig. S6), while it was 18 % for the entire period. An even larger effect of $CO_2$ fertilization was simulated in the extreme southern and northern parts of the study region (supplement S7 Fig. S6 C).

Mean total aboveground biomass averaged 66.4 T ha$^{-1}$ in eastern boreal Canada over the period 1901-2012. Mean total aboveground biomass decreased slightly from the beginning of the 20$^{th}$ C. until the 1930s, and then increased until 1995, after which it reached a stable level (Fig. 5 C). Periods of total aboveground biomass loss were recorded at the beginning of the 20$^{th}$ C. corresponded with high fire activity, as previously mentioned (Fig. 5 A). Sequential *t*-test analysis of total aboveground biomass time-series showed that biomass increase and reduction, respectively, followed the same trends that were observed for growth releases and declines until year 2000 (Fig. 6 C and D). Genus-specific aboveground biomass of *Picea*, *Pinus* and *Populus* PFTs showed the same increasing trends over the past century, whereas *Abies* PFT aboveground biomass decreased until year 1960 before regaining the value it had at the beginning of the 20$^{th}$ C (supplement S8 Fig. S7 A). The strongest variation in total aboveground biomass occurred for the *Picea* PFT, which varied from a minimum of 27.8 T ha$^{-1}$ in 1910 to a maximum of 36.7 T ha$^{-1}$ in 2003 (supplement S8 Fig. S7 A). Conversely, genus-specific aboveground biomass of *Abies*, *Pinus* and *Populus* PFTs varied by less than 1 T ha$^{-1}$, 2 T ha$^{-1}$ and 3 T ha$^{-1}$, respectively, over the same period (supplement S8 Fig. S7 A). The ratio of mean genus-specific aboveground biomass in the recent period of 1991-2012, when compared to the past period 1911-1930, was 1.23, 1.04, 1.13 and 1.31 for *Picea*, *Abies*, *Pinus* and *Populus* PFTs, respectively. The highest ratios for each PFT were found in the northern areas (supplement S8 Fig. S7 B).

## 4 Discussion

### 4.1 Agreements and disagreements in fire activity and forest growth

We used LPJ-LMfire, which was driven by gridded climatology, atmospheric [$CO_2$], and an estimate of lightning strike density, to study the pyrogeography of eastern Canada's boreal forest. Compared with previous modelling efforts that were conducted by Pfeiffer et al. (2013) using the original LPJ-LMfire model, the results that are reported here showed substantial improvements in the capacity of the DGVM to simulate fire ignition in the Canadian boreal forest. The use of a high-quality lightning strike dataset instead of the low-resolution LIS/OTD global dataset that was used by Pfeiffer et al. (2013) allowed us to capture the spatial gradient of fire activity in a substantially better manner (Baker et al., 2016). The results confirmed that fire in the study area is strongly ignition-limited, while most fire models have simply assumed that fire will always occur under appropriate weather and fuel conditions, e.g., SIMFIRE (Hantson et al. 2016). LPJ-LMfire simulations confirmed the necessity of simulating fire in a model as the product of the probabilities that are associated with fuel, moisture and ignition.

The inter-annual variation in lightning strike density was more faithfully reproduced when weighting the mean flash climatology with the CAPE variable to predict lightning-induced fire ignitions and their variability (Peterson et al., 2010). However, this variation is still constrained by the short temporal depth of the 11 years of record in the CLDN lightning strike dataset (Orville et al., 2002; Kochtubajda and Burrows, 2010). Synchronicity in major fire activity years across provinces (e.g.,

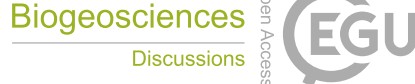

1961, 2005, 2007) was consistent with several studies on fire history, suggesting that changes in forest fire activity were observed conjointly over vast areas since the 1900s (e.g., Bergeron et al., 2004b; Macias Fauria and Johnson, 2008).

Annual burn rates (recent and historical) were underestimated in many areas of northern Quebec. It appears that the simulation could not capture the expression of a climate type that is encountered in the Clay Belt of northwestern Quebec, where periodic drought is known to occur. This likely may reflect some limitation that is imposed by the low density of weather stations north of 49°N. Lack of station replication can create excessively smoothed climate records, thereby reducing the possibility of correctly emulating the relationship between climate and forest fire activity during extreme drought and fire years (Girardin et al., 2006b, 2009; Xiao and Zhuang, 2007). For example, 1989 is known as a drought year, which was induced by changes in atmospheric circulation and that was at the origin of numerous large fires (> 50,000 ha) in Manitoba and Quebec (supplement S5 Fig. S4; Goetz et al., 2006; Xiao and Zhuang, 2007). Other large fires exceeding 50,000 ha were observed in northern Quebec in 1983 and 2002 (supplement S5 Fig. S4). However, these extreme weather conditions were not reproduced in our input dataset and, consequently, the model could not simulate these very large fires. These underestimates may also result, in part, from the lack of lightning strike records in these northernmost regions, which prevents fire ignition from being simulated there. Polarity or energy of lightning was not taken into account in our simulations. Positive lightning strikes (transfers of positive charges to the ground) mainly occur in the north and correspond to 10 % of all strikes (Morissette and Gauthier, 2008), with the remaining strikes being negative. Positive strikes correspond to an exchange of energy between the highest part of clouds and the soil, while negative strikes are triggered in a lower part of the cloud. For this reason, positive strikes are more likely to start fires because they carry higher energy owing to the greater travelling distance between clouds and the soil (Flannigan and Wotton, 1991). As previously mentioned, the number of lightning sensors in northern regions (hatched areas on Fig. 4) is also limited (Orville et al., 2011), leading to decreasing detection efficiency over these latitudes (Morissette and Gauthier, 2008). Thus, 10 % of positive strikes are not appropriately captured and, consequently, the probability of fire ignition also is likely to be underestimated in these areas. Underestimation of fire activity in northern areas had consequences for the simulation results. Amongst other things, simulated tree mortality was underestimated and, hence, biomass proliferated (as can be noted in Fig. 4 with *Picea* PFT).

Overestimates of simulated burn rates were reported for southern Ontario and Quebec. Indeed, some significant differences were expected with observed data in southern areas because they take into account anthropogenic effects. Forest management has contributed to modifying the composition and distribution of forest fuels (Girardin and Terrier, 2015; Danneyrolles et al., 2016). However, alterations in fuel composition and structure that were induced by human activities are not implemented in the present LPJ-LMfire simulations.

## 4.2 History of fire in the eastern boreal forest of Canada described by LPJ-LMfire

Based upon the above preliminary agreement and despite some disagreements, the temporal patterns of annual burn rates that were simulated by LPJ-LMfire were strongly consistent with forest fire histories that have been reconstructed in many studies (e.g., Stocks et al., 2003; Bergeron et al., 2004a; Girardin et al., 2006a). Multi-decadal temporal changes in annual burn rates




reflect the underlying influence of climate variability and extreme fire weather (Macias Fauria and Johnson, 2008; Girardin et al., 2009); these multi-decadal temporal changes were well represented in the input climate datasets. An increase in temperatures and stability of precipitation between 1916 and 1924 (supplement S9 Fig. S8) could be the origin of a high frequency of fire occurrence during those years, marking a pause in the decline of fire activity that had been observed since

the 1850s (Bergeron et al., 2004a). Advection of humid air masses over eastern Canada between 1940 and 1970 contributed to the creation of moister conditions (supplement S9 Fig. S8), which can lessen the capacity of a fire to spread after a lightning-induced fire ignition (Macias Fauria and Johnson, 2008). Both inter-annual variation and unsynchronized trends in climatic variables may have brought about changes in fire activity and could have affected the fire-season, as has been proposed to have occurred over millennial timescales during the Holocene (Ali et al., 2012). For example, during years 1977 and 1980, an

increase in spring temperatures was observed, whereas spring precipitation decreased, which resulted in the total areas that were burned in spring being 50 % greater than in summer (supplement S9 Fig. S8).

Correlations between simulated and observed provincial annual burn rates were slightly higher than have been typically encountered in past studies of fire-climate relationships over the region (e.g., Girardin et al., 2004, 2006a, 2009). For example, Girardin et al. (2009) reported that about 35 % of the variance in the annual areas that were burned in the provinces of Ontario

and Quebec was explained by summer moisture availability. In our modelling experiment, we obtained values between 41 % and 50 % for these same provinces, without empirical adjustments (e.g., through regression analysis). The improvements that were made here reinforce the idea that aside from "top-down" climate control on fire activity, other factors such as lightning, fuel availability and composition can influence fire statistics (Podur et al., 2002). This highlights the necessity of reconstructing fire history in a complex system that is related to climate and vegetation by taking into account several feedbacks (Hantson et

al., 2016). The strong correlations between our simulated annual burn rates and observed data further suggest that active fire suppression efforts and forest management since about the 1950s (Le Goff et al., 2008; Lefort et al., 2003) have not contributed much to shifting fire behaviour trajectories in our study region, which admittedly has very low densities of both population and infrastructure.

LPJ-LMfire correctly simulated the combination of fuel conditions and ignition sources to simulate fires. Indeed, an

increase in precipitation around the 1930s constrained fire activity, despite a very high lightning strike density (supplement S9 Fig. S8). Conversely, at the end of the century, an increase in strike density resulted in an increase of annual burn rates, in the presence of a relatively constant amount of precipitation (supplement S9 Fig. S8). Moreover, LPJ-LMfire does not simulate the core of the fire season between June and August when the highest density of lightning strikes takes place (Morissette and Gauthier, 2008). This result showed that heavy and intense rain events that occur later in the summer decrease the probability

of starting fires, despite more lightning.

Our simulation was biased with regard to fire seasonality. LPJ-LMfire simulated the core of the fire-season earlier than in what is actually observed. LMfire excludes fire ignition when snow cover is present (Pfeiffer et al., 2013). However, detailed investigations at the grid-cell level on our study area revealed that the Fire Danger Index, which was calculated by the LMfire module, was high as soon as all snow had melted in May and June. This index estimates the probability that an ignition event




will start a fire, depending upon both fuel moisture and fire weather conditions (Thonicke et al., 2010). As suggested by Pfeiffer et al. (2013), LPJ-LMfire simulates a very quick drying-out of soils in spring when the snow cover has disappeared or snowmelt has occurred prematurely. This phenomenon may be the reason why we were able to simulate the occurrence of fire-season onset earlier than what is observed in Canada's eastern boreal forest.

$CO_2$-induced enhancement of NPP (Norby et al., 2005; Huang et al., 2007) was clearly emulated in LPJ-LMfire. Our simulated 18 % growth enhancement with a 50 % increase in $[CO_2]$ between 1901 and 2012 was higher than the 15 % and 14 % growth increases that have been proposed by Hickler et al. (2008) and Girardin et al. (2011), respectively. LPJ-LMfire is highly sensitive to atmospheric $[CO_2]$ and interpreting its impacts must be made with caution (Girardin et al. 2011). That being said, our results suggest that $CO_2$-induced enhancement of forest productivity can be offset by fires and climate, which is
consistent with the results of Hayes et al. (2011) and Kelly et al. (2016). Despite strong $CO_2$-induced enhancement of forest productivity in LPJ-LMfire, the total amount of aboveground biomass and forest composition did not indeed change significantly during the course of the simulation period. Under very dry conditions, such as 1971-1990 and 1991-2012, an increase in fire activity led to a decrease in growth and biomass. Drier conditions during the past few decades provided indications for an increase in growth decline events and in biomass reduction that was related to an increase in fire activity. A
similar trend in such conditions was observed around the 1920s, but the range of these negative events during past decades exceeded the historical range of variability that was recorded by the simulated forest. Fires had a non-negligible influence on the state of the boreal forest in eastern Canada, especially during the last few decades, but our results also confirmed the relative influence of climate alone on the forest in northern regions. Indeed, in northern areas in Quebec and Manitoba, biomass has not significantly increased, despite a very strong effect of $CO_2$-induced enhancement (supplement S7 Fig. S6). We hypothesize
that with ongoing global warming, growth decline events could increase substantially, given that the positive effect of $CO_2$ on the growth of forest may not be strong enough to compensate for losses of biomass to fires and climate change (Kurz et al., 2008), which could lead to the opening up of landscapes.

## 4.3 Uncertainties and future perspectives

The present study has demonstrated that LPJ-LMfire is generally able to capture fire history and forest growth trends in the
eastern boreal forest of Canada. However, several uncertainties persist. First, forest establishment and the start of growth during the "spinup" phase was simulated using a detrended version of modern climate, as is usually performed in DGVM runs (Prentice et al., 2011; Pfeiffer et al., 2013; Yue et al., 2015; Knorr et al., 2016). This initial condition assumes that past relationships between climate, fire, and vegetation have been stationary through time and that variability of modern climate is representative of all variability that has been recorded over the past 1200 years (time of spinup phase + 112 years of simulation).
Yet, it has been increasingly recognized that such an assumption is invalid and that modern observations are not a good analogue for prehistoric variability (Kelly et al., 2016; Hudiburg et al., 2017). For example, fire activity over much of the Holocene was higher in terms of frequency and fire sizes than are current levels across broad areas of eastern Canada (Girardin et al., 2013a; Remy et al., 2017). It is likely that not accounting for such variability may introduce biases in forest productivity





dynamics and levels, more specifically on soil carbon dynamics (Hudiburg et al., 2017). This may be less problematic when studying fire and forest dynamics over the last century because the mean age of the major part of eastern boreal forest is less than 100 years (Bergeron et al., 2002).

The non-negligible influence of forest composition on fire regimes (Hély et al., 2001) is limited in the model to the
representation of three needle-leaf PFTs and one broad-leaf PFT. Improving LPJ-LMfire's representation of biodiversity with further broad-leaf PFT genera could counterbalance or offset overestimates of fire activity in southern areas since these species are less flammable than needle-leaf species. Similarly, improving LPJ-LMfire parametrization to account for mosses could reduce overestimation of quantities of fuel that was available in northern areas. In the Clay Belt, poor drainage conditions that were induced by the presence of an impermeable clay substrate, flat topography and a cold climate facilitate the accumulation
of thick layers of organic soil, an edaphic process that is described as paludification (Fenton et al., 2006). Once *Sphagnum* species increase on the forest floor, the depth of burn varies only slightly in response to changes in weather conditions, owing to very low fluctuations in the degree of water saturation of the organic layer (Fenton et al., 2006).

In the present study, simulations are limited by the relatively low accuracy of soil attributes in databases for Canada's boreal forest (Hengl et al., 2014). The input dataset of soil attributes that was used in our simulations tended to underestimate clay
and sand percentages on our study area when compared to point observations (supplement S10 Fig. S9). These effects add-up to other weaknesses in the physiological constraints, such as cold climate not being sufficiently restrictive and allowing *Picea* to become overly abundant in the simulation runs. While a previous study showed that the abundance of *Picea* decreases with latitude in the Tundra region and is coupled with the occurrence of dwarf shrubs in the *Ericaceae* and herbs (Gajewski et al., 1993), such species were not parametrized in the current version of LPJ-LMfire, due to a lack of information on their
physiological and biogeographical preferences. Future research could incorporate recently developed parameterizations for boreal shrubs and non-vascular plants into LPJ-LMfire (Druel, 2017; Druel et al., 2017).

Forest stand structure and successional dynamics (age classes), together with processes leading to the formation of peatlands, are not included in the present version of LPJ-LMfire. Yet, all of these aspects are important determinants of fire ignition and propagation under a given climate (Hély et al., 2001) and also can influence the distinction between crown and
surface fires, which affect tree mortality differently (Hély et al., 2003; Yue et al., 2015). Moreover, LPJ-LMfire, like most DGVMs, does not consider constraints on species migrations, phenotypic plasticity, and local adaptation of species (Morin and Thuiller, 2009). The simulation results may be overly optimistic in terms of the capacity of southern species to colonize newly available areas in northern regions as the climate warms. As previously mentioned by Morin and Thuiller (2009), species colonization in northern regions could be limited by forest attributes, such as fragmented landscapes or high competition levels
from existing species, or through migrational lag (Epstein et al., 2007).

Wildland fires are the most important natural disturbances in Canada's eastern boreal forest, but non-fire disturbances also have considerable effects (Price et al., 2013) and may influence fire activity trajectories indirectly. Integrating a range of forest disturbances into a DGVM could improve the accuracy of forecasting and modelling of climate change effects on Canada's eastern boreal forest. For instance, insect damage (MacLean, 2016), and outbreaks of eastern spruce budworm (*Choristoneura*

*fumiferana*) in particular (Zhang et al., 2014, James et al., 2017), represent a significant forest disturbance by temporarily altering forest structure by affecting specific tree growth, tree survival, regeneration and succession. These disturbances can have an important impact on fire activity, in turn, by modifying fuel distribution and connectivity (James et al., 2017). Additionally, successive fires that take place over a short period before the trees have attained maturity have led to complete regeneration failure (Girard et al., 2008). Such events in young, unproductive stands can also lead to modified forest composition (Girard et al., 2008) and could exert a strong feedback on ecosystem structure by generating changes in temporal fire patterns on long timescales.

## 5. Conclusions

In this study, we used LPJ-LMfire to simulate fire activity from 1901 to 2012 in Canada's eastern boreal forest, at 10 km resolution. LPJ-LMfire was parametrized for the predominant forest tree genera that were present in our study region, viz., *Picea*, *Abies*, *Pinus* and *Populus*. The predictive skill of the model to simulate fire activity was determined by comparing our model simulations with published data. LPJ-LMfire was able to simulate inter-annual- to decadal-scale fire variability from the beginning of the 20th C. However, the low density of weather stations in northern areas likely limited the model's ability to capture some extreme fire years. Our study highlights the importance of changes in climate variables at multi-decadal and annual timescales in strongly controlling spatiotemporal patterns of fire that were simulated by LPJ-LMfire. Spatiotemporal patterns were well captured, based upon our climate data inputs. Despite an overarching $CO_2$-induced enhancement of NPP in LPJ-LMfire, aboveground biomass was relatively stable because of the compensatory effects of increasing fire activity. This study helps reduce uncertainties in our knowledge regarding fire patterns in the recent past and confirms that fires were a dominant driver of boreal forest in eastern Canada during the last century. We further provide a new tool to refine predictions of future fire risks and effects of ongoing climate change in these forests to better inform management and improve risk mitigation strategies.

## Code availability

The source code of LPJ-LMfire is available at: https://github.com/ARVE-Research/LPJ-LMfire/tree/canada/src.

## Author contributors

E. Chaste, M. Girardin, Y. Bergeron and C. Hély conceived and designed the study. E. Chaste performed the simulations and preparation of input datasets with the help of J. Kaplan. J. Portier performed statistical calculations of annual burn rates for our simulation period according to the protocol that is described by Portier et al. (2016). E. Chaste, M. Girardin, J. Kaplan, Y. Bergeron and C. Hély interpreted the results. E C. Chaste prepared the manuscript with contributions from all co-authors.





**Competing financial interests**

The authors declare that they have no conflict of interest.

**Acknowledgements**

The study was made possible thanks to financial support provided by the European IRSES NEWFOREST project, the Forest

Complexity Modelling (FCM) program and the NSERC Strategic and Discovery programs. Jed Kaplan was supported by the

European Research Council (COEVOLVE 313797). This research was conducted as part of the International Associated

Laboratory MONTABOR (LIA France-Canada) and the International Research Group on Cold Forests. We thank Melanie

Desrochers and Xiao Jing Guo for their help with mapping and computation for this project. We also thank Daniel Stubbs from

Calcul Quebec and Compute Canada for the Fortran language help and server space facilities for running LPJ-LMfire. We also

thank W.F.J. Parsons for English-language editing of the draft of this manuscript.

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





**Captions**

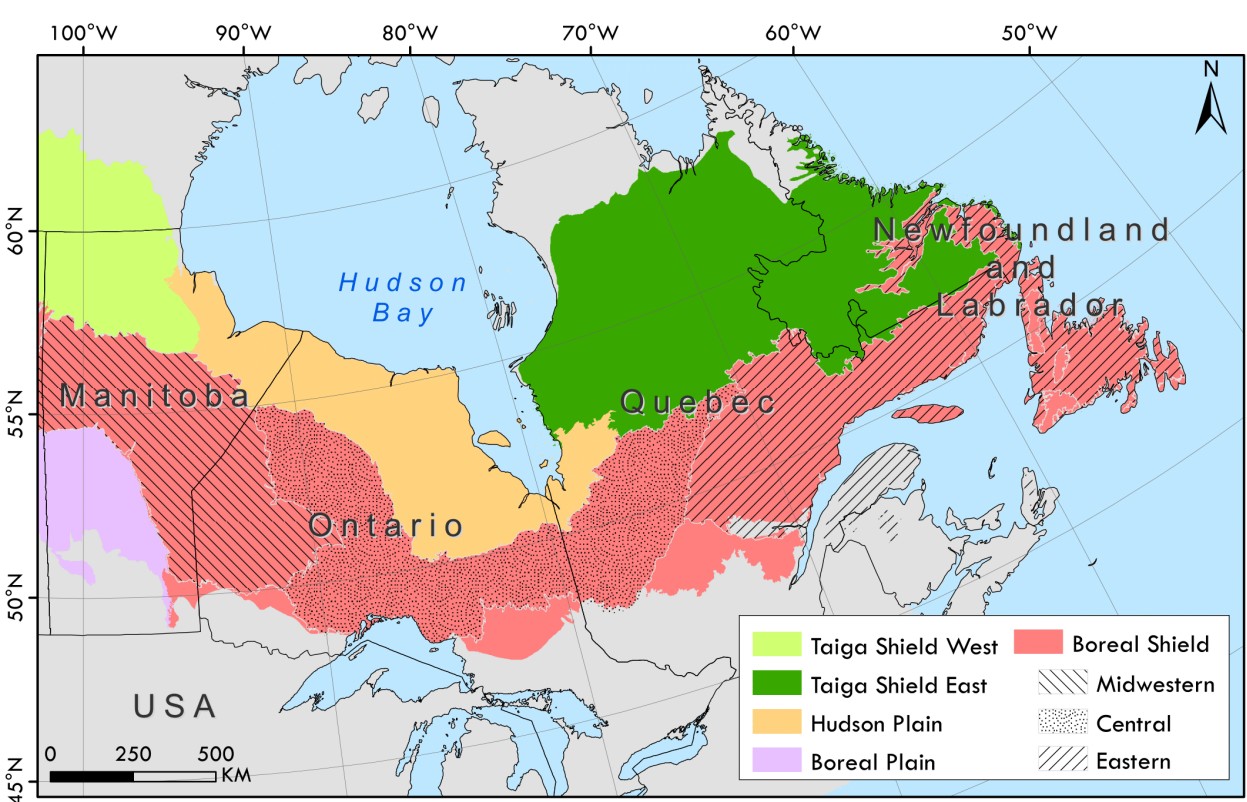

**Figure 1: Map of eastern Canada's boreal forest from Manitoba to Newfoundland showing ecozones in colour. The ecozone "Boreal Shield" is divided into three ecoregions: eastern Canadian forests, central Canadian Shield forests and midwestern Canadian Shield forests.**



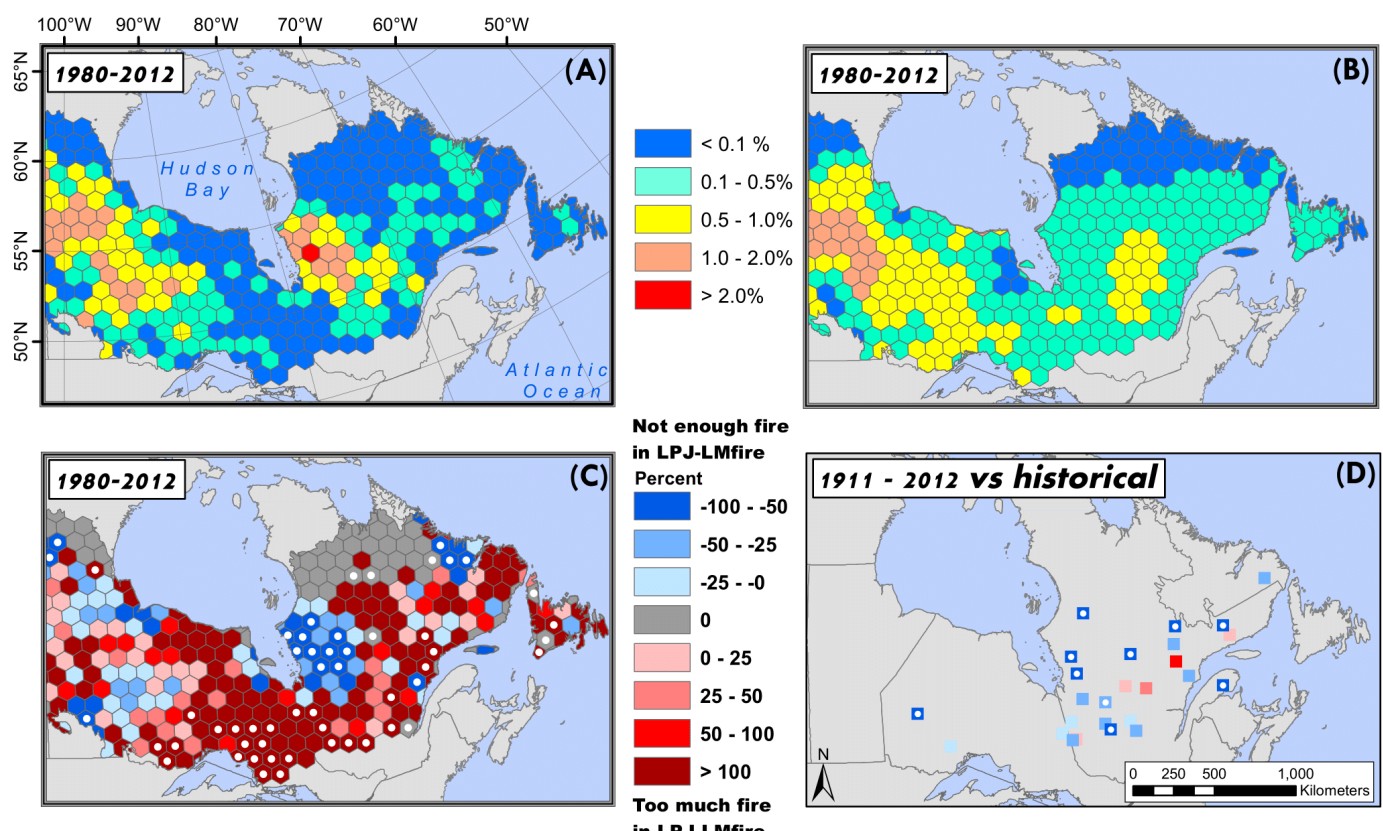

**Figure 2: Observed versus LPJ-LMfire simulated annual burn rates across eastern boreal Canada. (A) Observed annual burn rates computed for 365 hexagonal cells between 1980 and 2012 (data from Natural Resources Canada, 2017). (B) LPJ-LMfire simulated annual burn rates computed over the same period and hexagonal cells. (C) Percent difference between observed and simulated annual burn rates. (D) Percent difference between historical annual burn rates reconstructed from stand-replacing fire history studies (data from Girardin et al., 2013b; Héon et al., 2014; Portier et al., 2016) and LPJ-LMfire simulated annual burn rates between 1911 to 2012 (see supplement S4 for further details). White points indicate where the observed (and historical) annual burn rate lies outside the 95 % confidence interval (95 % CI) of the averaged annual burn rates in hexagonal cell simulated by LPJ-LMfire.**



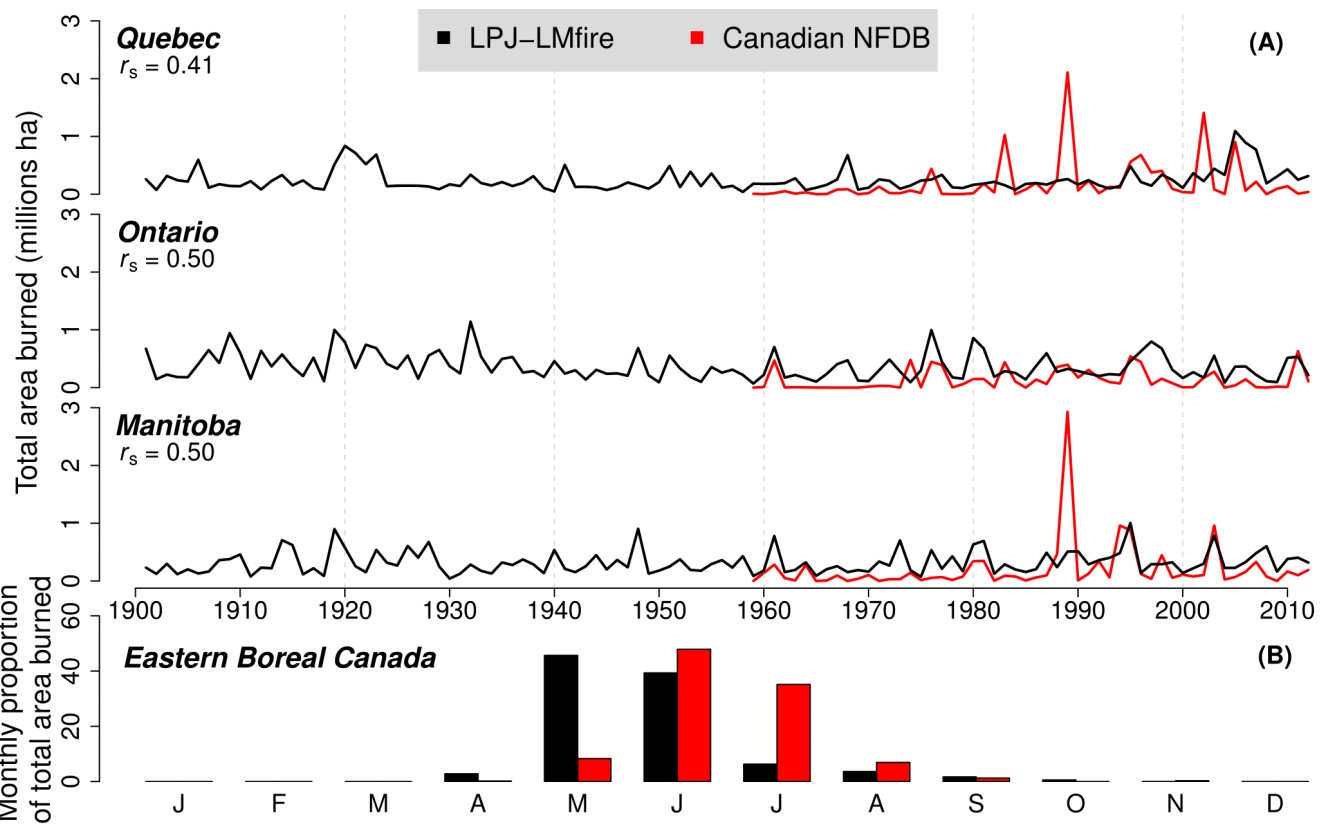

**Figure 3: (A) Observed versus simulated annual total areas burned in three provinces of eastern Canada. Observed data (1959 to 2012) are from the Canadian National Fire Database (CNFDB). The Spearman rank correlation between data is shown (correlations are significant at P < 0.05 for Quebec and P < 0.001 for the other provinces). (B) Monthly percentage of total areas burned between 1959 and 2012 in eastern boreal Canada.**





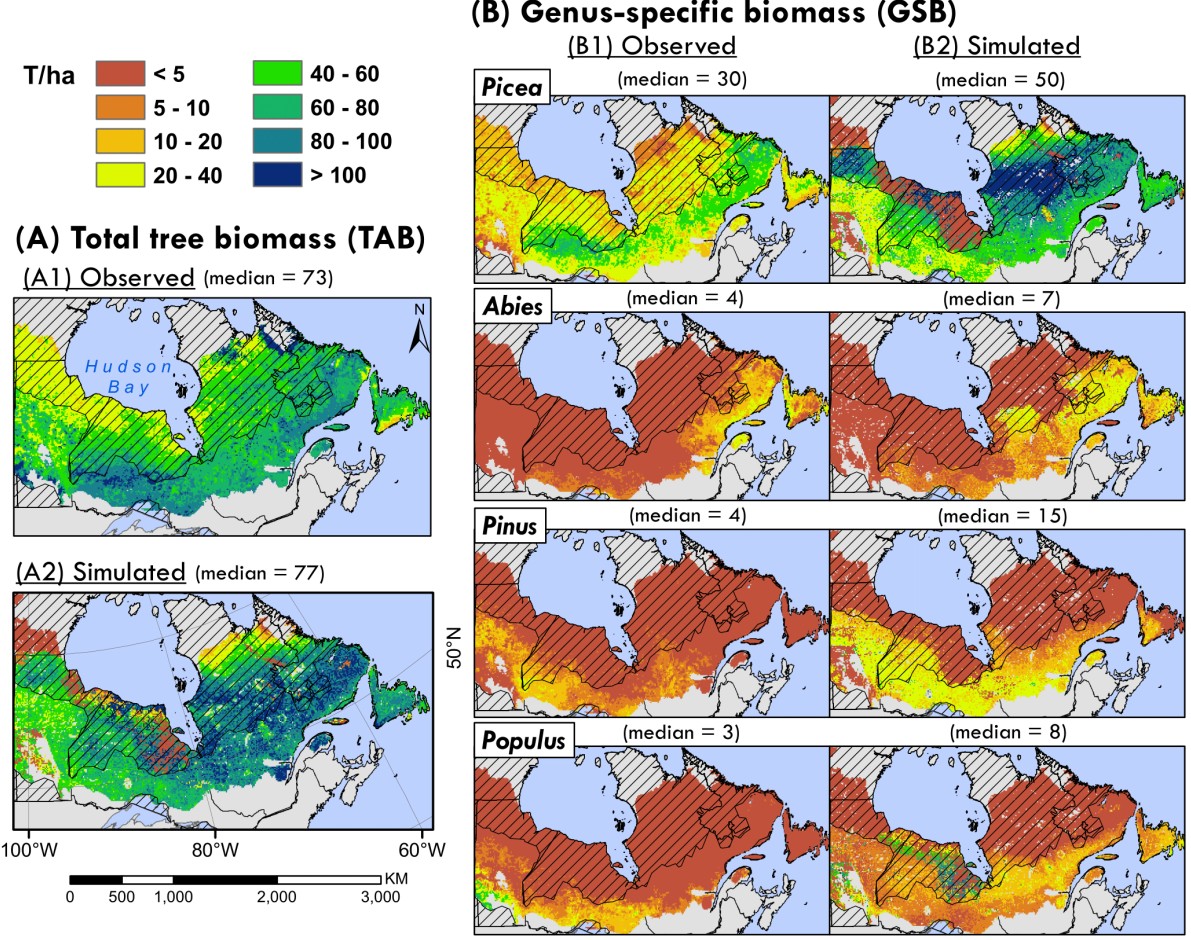

**Figure 4: (A) Observed versus LPJ-LMfire simulated mean total aboveground biomass (T/ha) between 2000 and 2006 across eastern boreal Canada. (B) Genus-specific aboveground biomass. The observed aboveground biomass maps across Canada were predicted and validated with photo-plot information in the southern areas (non-hatched areas) and data published by Beaudoin et al. (2014). Median tree aboveground biomass values were calculated for the non-hatched area.**



**Figure 5: LPJ-LMfire simulated (A) annual burn rates (%), (B) net primary productivity (T/ha/yr), and (C) total aboveground biomass (T/ha) across eastern boreal Canada for five periods between 1911 and 2012.**



**Figure 6:** Annual (A) and decadal (smoothed 10-year sums; B) proportions of cells showing a significant decline and release in NPP with 90 % confidence intervals (error bars) computed using Bayes method. Annual (C) and decadal (smoothed 10-year sums; D) proportions of cells showing a significant reduction and increase in biomass total aboveground with 90 % confidence intervals (error bars) computed using the same method.



**Tables**

**Table 1: Climate and other data sets used to drive LPJ-LMfire.**

| Variables (units) | Period | Datasets | References |
|---|---|---|---|
| Monthly mean temperature (°C) and monthly mean diurnal temperature range (°C) | 1901-2012 | | |
| Monthly mean precipitation (mm) and number of days per month with precipitation | | | |
| Monthly mean of wind speed (m.s$^{-1}$) | 1969-2010 | | |
| Monthly mean of total cloud cover percentage | | | |
| Monthly mean convectible available potential energy (J.kg$^{-1}$) | | | |
| Lightning flashes (number/day/km$^2$) | 1999-2010 | Canadian lightning detection network | (Orville et al., 2011) |
| Soil particle size distribution and volume fraction of coarse fragments (%) | - | ISRIC - World Soil Information | (Hengl et al., 2014) |
| Elevation (meters) and slope (degrees) | - | Canada 3D | (Natural Resources Canada, 2007) |
| Water fraction | - | National Hydro Network (NHN) | (Natural Resources Canada, 2010) |
| Atmospheric $CO_2$ concentrations (ppm) | - | Composite $CO_2$ time-series | (Keeling et al., 2009; Pfeiffer et al., 2013) |





**Table 2: LPJ-LMfire vs Margolis et al. (2015) mean total aboveground biomass estimates (with standard deviations) between 2000 and 2006 across 5 ecozones in eastern boreal Canada. The Boreal Shield ecozone was divided into three ecoregions (ecozone subdivisions) for comparison.**

| Zone | Ecozones | Ecoregions | Mean TAB (T.ha$^{-1}$) | | | |
|---|---|---|---|---|---|---|
| | | | LPJ-LMfire | GLAS | NFI | kNN |
| | Taiga shield east | - | 72.8 (30.0) | 44.5 | | 39.8 |
| | Taiga shield west | - | 38.6 (33.2) | 38.1 | | 25 |
| | Hudson plain | - | 59.0 (43.1) | 26.1 | 24.4 | 37.2 |
| | | Eastern Canadian forests | 88.7 (17.7) | 67.9 | | 64.7 |
| | | Central Canadian Shield forests | 78.8 (17.3) | 68.4 | | 67.8 |
| | | Midwestern Canadian Shield forests | 57.6 (15.1) | 56.4 | | 52.8 |
| | Boreal plain | - | 31.9 (23.5) | 64.0 | 79.9 | 55.5 |

