# Peer review of "The pyrogeography of eastern boreal Canada from 1901 to 2012 simulated with the LPJ-LMfire model"

_Biogeosciences, 2017_

## Referee Comment (RC1) · Anonymous Referee #1 · 30 Oct 2017

General comments: This paper addressed the ability of the dynamic global vegetation model LPJ-LMFire to accurately represent past fire activity in the boreal forests of eastern Canada. They found that increases in NPP due to increasing $CO_2$ concentration can be offset by increases in fire activity in more northern areas. I interpret their results and discussion to say that the increase in fore activity was due to both an increase in lightning frequency and drier climate. I recommend emphasizing this aspect of their results more clearly and explicitly.. This paper provides both methodology and results that will be of interest to the community.

Specific Comments: 1. It appears that there were no additional modifications made to

the fire routine in LPJ_LMfire, such as changes to fuel limits on fire activity, beyond the new PFT parameterization. If true, I would encourage the authors to write a sentence confirming this.

1. Presenting the results of Figure 4 in a manner similar to Figure 2 would make it easier for the reader to see the spatial patterns of agreement and disagreement between model and observations.

2. Given the results shown in Figure 3, the interpretation that "heavy and intense rain events that occur later in the summer decrease the probability of starting fires, despite more lightning" does not seem well-founded.

Technical Errors: 1. There are multiple instances of incorrect grammar.
* * *

---

## Author Comment (AC1) · 16 Nov 2017

Dear Referee #1,

Thank you very much for your comments and suggestions for improvement and clarification of the manuscript. Here are individual responses to your comments and some details about the revisions we plan to make for the final acceptance:

- Referee #1 comment #1: I interpret their results and discussion to say that the increase in fire activity was due to both an increase in lightning frequency and drier climate. I recommend emphasizing this aspect of their results more clearly and explicitly.

Authors #1: We agree with you and we will change one sentence in the abstract and one paragraph in the discussion section to emphasize that the increase in fire activity during the last part of the 20th century was the result of both an increase in lightning frequency and a drier climate. This aspect was mentioned in the 3rd paragraph, section "4.2 History of fire in the eastern boreal forest of Canada described by LPJ-LMfire" but it was perhaps not sufficiently emphasized. We will highlight it better in the final version.

- Referee #1 comment #2: It appears that there were no additional modifications made to the fire routine in LPJ_LMfire, such as changes to fuel limits on fire activity, beyond the new PFT parameterization. If true, I would encourage the authors to write a sentence confirming this. Authors #2: No modification was made to the fire routine compared to the version of Pfeiffer et al. (2013); we will make sure that is explicitly mentioned in the Method section of the final version.

- Referee #1 comment #3: Presenting the results of Figure 4 in a manner similar to Figure 2 would make it easier for the reader to see the spatial patterns of agreement and disagreement between model and observations. Authors #3: We really appreciate this comment to improve clarity and uniformity between all figures related to the part "3.1 Predictive skills of the LPJ-LMfire model". As suggested, we will present the results of Figure 4 in the format of Figure 2 (see Fig. 1 below). Maps reporting differences between mean aboveground biomass simulated and observed were added on the Figure 4.

- Referee #1 comment #4: Given the results shown in Figure 3, the interpretation that "heavy and intense rain events that occur later in the summer decrease the probability of starting fires, despite more lightning" does not seem well-founded. Authors #4: We agree with you concerning this sentence and we think that, if LPJ-LMfire does not simulate the bulk of the fire season between June and August, it is mainly due to a quick drying-out of soils in spring within the simulation. Here, we just suggest that the low burn rate during the summer despite high lightning density (see Fig. 2 below)

could also result from the weather being less conducive to fire due to higher amount of precipitation between July and September than between April to June (supplement S9 Fig. S8). Detailed investigations (not shown in the manuscript) at the grid-cell level have highlighted that the Fire Danger Index calculated by LPJ-LMfire was higher in spring than in summer. This index is directly related to the amount of precipitation that influences fuel moisture. We will improve our sentence both to clearly explain this point and to make sure readers take this information as a suggested interpretation.

- Referee #1 comment #5: There are multiple instances of incorrect grammar. Authors #5: The final version will be reedited for grammar and spelling.

Thank you for reviewing our paper.

Sincerely yours, Emeline Chaste and coauthors

—————————————————

[Figure]

Figure 4: (A) Observed, LPJ-LMfire simulated, and differences (%) in mean total aboveground biomass (T/ha) between 2000 and 2006 across eastern boreal Canada. (B) Genus-specific aboveground biomasses. The observed aboveground biomass maps across Canada were predicted and validated with photo-plot information in the southern areas (non-hatched areas) and data published by Beaudoin et al. (2014). Median (m) tree aboveground biomass values were also mentioned for each maps and were calculated for the non-hatched area at 10 km resolution.

**Fig. 1.**

[Figure]

Figure 1: Temporal series of spring and summer mean flashes density.

**Fig. 2.**

---

## Referee Comment (RC2) · Anonymous Referee #2 · 14 Dec 2017

General comments: The goal of this paper was to reconstruct the fire activity of the boreal forest in eastern Canada using the LPJ-LMFIRE dynamic vegetation model. Additional goals were to calibrate and validate the model and to determine how spatiotemporal patterns in fire activity, NPP, and aboveground biomass have interacted with climate over the past century. They found that the ability of the LPJ-LMFIRE model to capture spatial patterns in fire activity was significantly improved through incorporation of a high-quality lightning dataset. Overall, the model performed well at reproducing pattern in fire frequency, but with some area of under- and overestimation. Additionally, the authors conclude that $CO_2$ fertilization has contributed to increases in NPP, but that increased fire activity offsets this increase leading to relatively stable aboveground

tree biomass. I found the paper to be well written and appreciated the thorough treatment of model validation. This paper fits well within the scope of Biogeosciences and represents an important contribution to fire and vegetation simulation studies.

Specific comments: 1. I would like to see a more thorough treatment of the differences between the "Climate + CO2" and "Climate only" simulations, specifically with respect to how fire activity differed between the two scenarios. It was not clear to me whether the fire activity results reported in section "3.2.1 Fire activity" and Fig. 5 were from the "Climate only" simulation or the "Climate+CO2" simulations. Because the authors emphasize that CO2 fertilization increases NPP but that this is offset by increased fire activity, I think the results for fire activity with and without CO2 need to be more clearly presented. The effects of CO2 on NPP are discussed in section 3.2.2. Fuels, but I do not see a similar discussion of the fire results.

2. Section 2.2 LPJ-LMfire model: Does the LPJ-LMfire model allow for cell to cell spread of fire? It is my understanding that the base LPJ model has no cell to cell interaction, a major limitation for simulating fire. It was my understanding that the SPITFIRE model did incorporate cell-cell interactions, but this needs to be explicitly stated in the manuscript.

3. Study area description: I think it would benefit the reader if the authors included some rough description of fire regimes (general fire frequencies and whether historical fires were predominantly low, mixed, or high-severity, for example) for each of the ecoregions. As written, only the Boreal Shield ecoregions is characterized as having "typically high" fire frequency, but no estimate is given as to what that means in terms of mean fire return interval.

4. On p14 L20-23, the authors suggest that strong correlations between simulated and observed annual burn rates indicate fire suppression efforts have had little affect on fire behavior. Although this makes sense overall, the authors also note on p13 lines 25-30 that overestimates of simulated burn rates in Ontario and Quebec were expected because the model doesn't consider anthropogenic effects and that forest management has influenced forest composition/fuels. Perhaps these paragraphs need to be combined and the nuances of how fire and forest management have varied spatially be expanded upon?

5. Table 1. The datasets and references for the temperature, precipitation, cloud cover and convectible available potential energy were listed in the text, but I found it odd for them not be listed in the table when that information is listed in the table for the remaining variables.

6. P3 Lines 11-16. I find it odd that the authors claim a "lack of quantitative knowledge regarding the relative changes in fire regimes that are attributable to human impacts". Perhaps this is true of boreal Canada, I am more familiar with fire history studies in the U.S. Southwest and US Pacific northwest. Especially in the U.S. Southwest, fire history and forest age/structure and species composition from dendroecological evidence have been used to quantify how human activities (logging, grazing, fire exclusion) have impacted fire regimes (see studies by Thomas Swetnam, Peter Fule, Peter Brown, among others). These statements need to be qualified.

Technical comments:

P1 Line 33-34: Change the sentence to read: "Agreement between the spatiotemporal patterns of fire frequency and the observed data confirmed that fire in the study are is strongly ignition-limited."

---

## Author Response (AR1)

**January 22, 2018**

**Reference:** Submission bg-2017-350

**Dear Dr. Stoy,**

We thank the two anonymous reviewers for providing recommendations and appreciate the time and effort they put into this study. We have made all changes requested by the referees.

Changes made to the text and responses to comments are in detail discussed below. Changes made to our manuscript include:

- Improvement of Figure 4
- Modification of the 3rd paragraph in the Introduction section (starting on page 3, line 12)
- Added details about the study area and methods
- Reorganization of several discussion subsections
- Addition in the Supplementary Material file of a new figure (Fig. S7 in Supplement S7) with annual burn rates
- Editorial revision for grammar and spelling performed by the federal editorial service of Natural Resources Canada

A version of the manuscript with tracked changes is enclosed with this submission.

We thank you for giving us the opportunity to improve our manuscript.

Sincerely yours,

Emeline Chaste and coauthors

*Referee #1 comment #1:* I interpret their results and discussion to say that the increase in fire activity was due to both an increase in lightning frequency and drier climate. I recommend emphasizing this aspect of their results more clearly and explicitly.

> *Authors #1:* We changed one sentence in the abstract and sentences in the 3$^{rd}$ paragraph, section "4.2 History of fire in the eastern boreal forest of Canada described by LPJ-LMfire" to improve this aspect.

*Referee #1 comment #2:* It appears that there were no additional modifications made to the fire routine in LPJ-LMfire, such as changes to fuel limits on fire activity, beyond the new PFT parameterization. If true, I would encourage the authors to write a sentence confirming this.

> *Authors #2:* We added the sentence "No other modifications were made to the Pfeiffer et al. (2013) version of LPJ-LMfire." at the end of the section "2.5 PFT definitions and LPJ-LMfire model modifications".

*Referee #1 comment #3:* Presenting the results of Figure 4 in a manner similar to Figure 2 would make it easier for the reader to see the spatial patterns of agreement and disagreement between model and observations.

> *Authors #3:* We presented the results of Figure 4 in the format of Figure 2. Maps reporting differences between mean aboveground biomass simulated and observed were added on the Figure 4.

*Referee #1 comment #4:* Given the results shown in Figure 3, the interpretation that "heavy and intense rain events that occur later in the summer decrease the probability of starting fires, despite more lightning" does not seem well-founded.

> *Authors #4:* We rewrote the 3$^{rd}$ paragraph, section "4.2 History of fire in the eastern boreal forest of Canada described by LPJ-LMfire" to improve the idea that both fuel conditions and ignition sources have an influence on annual burn rates simulated by LPJ-LMfire. Heavy and intense rain events that occur later in summer increase the moisture of fuel and consequently decrease the probability of starting fires despite high density of lightning strikes.

*Referee #1 comment #5:* There are multiple instances of incorrect grammar.

> *Authors #5:* The final version was reedited for grammar and spelling by a technical editor.

*Referee #2 comment #1:* I would like to see a more thorough treatment of the differences between the "Climate + CO2" and "Climate only" simulations, specifically with respect to how fire activity differed between the two scenarios. It was not clear to me whether the fire activity results reported in section "3.2.1 Fire activity" and Fig. 5 were from the "Climate only" simulation or the "Climate+C02" simulations. Because the authors emphasize that CO2 fertilization increases NPP but that this is offset by increased fire activity, I think the results for fire activity with and without C02 need to be more clearly presented. The effects of C02 on NPP are discussed in section 3.2.2. Fuels, but I do not see a similar discussion of the fire results.

*Authors #6:* All figures in the manuscript were from the 'Climate + CO$_2$' experiment; this aspect is now more clearly stated in the manuscript. We compared the NPP simulated by LPJ-LMfire with and without the CO$_2$ effect in supplement S7 Fig. S6 and explained the results in the 1$^{st}$ paragraph of the section "3.2.2 fuels". We also added in supplementary materials a new figure with a comparison of annual burn rates simulated by LPJ-LMfire for both 'Climate-only' and 'Climate + CO$_2$' experiments (Fig. S7 in Supplement S7). Results from this comparison were explained in section "3.2.1". In the discussion, we specified that the CO$_2$-induced enhancement of NPP had an influence on the annual burn rates by increasing the availability of fuel.

*Referee #2 comment #2:* Section 2.2 LPJ-LMfire model: Does the LPJ-LMfire model allow for cell to cell spread of fire? It is my understanding that the base LPJ model has no cell to cell interaction, a major limitation for simulating fire. It was my understanding that the SPITFIRE model did incorporate cell-cell interactions, but this needs to be explicitly stated in the manuscript.

*Authors #7:* No fire model currently used to simulate fire over continental to global-scale domains includes a representation of cell-to-cell fire spread (Pfeiffer et al., 2013; Hantson et al., 2016; Rabin et al., 2017). The basic concept in global fire modeling is that most fires are not as large as a single gridcell and so it is possible to imagine that an individual fire event occurs only within the size of an individual gridcell. We added the sentence "As in the original version of SPITFIRE and nearly all other large-scale fire models, LMfire does not simulate the cell-to-cell spread of fire (Hantson et al., 2016; Pfeiffer et al., 2013; Rabin et al., 2017)." at the end of the section "2.2 LPJ-LMfire model".

*Referee #2 comment #3:* Study area description: I think it would benefit the reader if the authors included some rough description of fire regimes (general fire frequencies and whether historical fires were predominantly low, mixed, or high-severity, for example) for each of the ecoregions. As written, only the Boreal Shield ecoregions is characterized as having "typically high" fire frequency, but no estimate is given as to what that means in terms of mean fire return interval.

*Authors #8:* We added some information about the fire regimes for each ecozones in section "2.1 Study area": Within the study area, high-intensity crown fires are the most common type of fire events (Flannigan et al., 2016). Fire regimes are heterogeneous, but generally follow a declining trend along a southwest-northeast gradient (Boulanger et al., 2012). During the period 1961–1990, the highest burn rates occurred in the western part of the BS ecozone (> 1% yr$^{-1}$), while they were the lowest in the TSE ecozone (< 0.2% yr$^{-1}$) (Boulanger et al., 2014). Annual burn rates in the BP ecozone and in the eastern part of the BS ecozone varied from 0.2 to 0.5% yr$^{-1}$, whereas it varied from 0.5 to 1.0% yr$^{-1}$ in the HP ecozone (Boulanger et al., 2014).

*Referee #2 comment #4:* On p14 L20-23, the authors suggest that strong correlations between simulated and observed annual burn rates indicate fire suppression efforts have had little affect on fire behavior. Although this makes sense overall, the authors also note on p13 lines 25-30 that overestimates of simulated burn rates in Ontario and Quebec were expected because the model doesn't consider anthropogenic effects and that forest management has influenced forest

composition/fuels. Perhaps these paragraphs need to be combined and the nuances of how fire and forest management have varied spatially be expanded upon?

> *Authors #9:* As proposed we combined these paragraphs at the end of the section "4.3 Uncertainties and future perspectives" and nuanced these sentences to clarify that the effect of humans on management and suppression is not taken into account in LPJ-LMfire. However, similar temporal patterns of simulated and observed annual burn rates showed there does not seem to be a detectable effect of human activities on fire history.

*Referee #2 comment #5:* Table 1. The datasets and references for the temperature, precipitation, cloud cover and convectible available potential energy were listed in the text, but I found it odd for them not be listed in the table when that information is listed in the table for the remaining variables.

> *Authors #10:* There was a problem during the conversion process to PDF and missing data were present in Table 1 and 2 of the submitted version. We will make sure that this problem does not happen again for the final version.

*Referee #2 comment #6:* P3 Lines 11-16. I find it odd that the authors claim a "lack of quantitative knowledge regarding the relative changes in fire regimes that are attributable to human impacts". Perhaps this is true of boreal Canada, I am more familiar with fire history studies in the U.S. Southwest and US Pacific northwest. Especially in the U.S. Southwest, fire history and forest age/structure and species composition from dendroecological evidence have been used to quantify how human activities (logging, grazing, fire exclusion) have impacted fire regimes (see studies by Thomas Swetnam, Peter Fule, Peter Brown, among others). These statements need to be qualified.

> *Authors #11:* We clarified the 3rd paragraph in the section "Introduction".

*Referee #2 comment #7:* P1 Line 33-34: Change the sentence to read: "Agreement between the spatiotemporal patterns of fire frequency and the observed data confirmed that fire in the study are is strongly ignition-limited."

> *Authors #12:* We changed the sentence in P1 line 33-34 as proposed.

**References**

[revised manuscript text omitted]